# Comparison of an in-house 'home-brew' and commercial ViroSeq integrase genotyping assays on HIV-1 subtype C samples

Kaelo K. Seatla[1,2], Wonderful T. Choga[3], Mompati Mogwele[1,2], Thabo Diphoko[1,2], Dorcas Maruapula[1,2], Lucy Mupfumi[1,2], Rosemary M. Musonda[1,4], Christopher F. Rowley[4], Ava Avalos[1,5,6], Ishmael Kasvosve[2], Sikhulile Moyo[1,4], Simani Gaseitsiwe[1,4]*

1 Botswana Harvard AIDS Institute Partnership Gaborone, Botswana, 2 Department of Medical Laboratory Sciences, School of Allied Health Professionals, University of Botswana, Gaborone, Botswana, 3 Division of Human Genetics, Department of Pathology, Faculty of Health Sciences, University of Cape Town, Cape Town, South Africa, 4 Department of Immunology & Infectious Diseases, Harvard T.H. Chan School of Public Health, Boston, Massachusetts, United States of America, 5 Careena Centre for Health, Gaborone, Botswana, 6 Ministry of Health and Wellness, Gaborone, Botswana

* sgaseitsiwe@gmail.com

**Data Availability Statement:** The 33 pairs of nucleotide sequences obtained in our study were submitted to National Center for Biotechnology

## Abstract

### Background

Roll-out of Integrase Strand Transfer Inhibitors (INSTIs) such as dolutegravir for HIV combination antiretroviral therapy (cART) in sub-Saharan Africa necessitates the development of affordable HIV drug resistance (HIVDR) assays targeting the Integrase gene. We optimised and evaluated an in-house integrase HIV-1 drug resistance assay (IH-Int) and compared it to a commercially available assay, ViroSeq™ Integrase Genotyping kit (VS-Int) amongst HIV-1 clade C infected individuals.

### Methods

We used 54 plasma samples from treatment naïve participants and one plasma sample from a patient failing INSTI based cART. Specimens were genotyped using both the VS-Int and IH-Int assays. Stanford HIV drug resistance database were used for integrase resistance interpretation. We compared the major and minor resistance mutations, pairwise nucleotide and amino-acid identity, costs and assay time.

### Results

Among 55 specimens tested with IH-Int, 53 (96.4%) successfully amplified compared to 45/55 (81.8%) for the VS-Int assay. The mean nucleotide and amino acid similarity from 33 paired sequences was 99.8% (SD ± 0.30) and 99.8% (SD ± 0.39) for the IH-Int and VS-Int assay respectively. The reagent cost/sample were 32 USD and 147 USD for IH-Int and VS-Int assay, respectively. All sequenced samples were confirmed as HIV-1 subtype C.

Information (NCBI) GenBank and the accession numbers are MN037428 to MN037493. Additional nucleotide sequences from our study are available in GenBank under accession numbers MN462669 to MN462690.

**Funding:** This work was supported through the Sub-Saharan African Network for TB/HIV Research Excellence (SANTHE), a DELTAS Africa Initiative [grant # DEL-15-006]. The DELTAS Africa Initiative is an independent funding scheme of the African Academy of Sciences (AAS)'s Alliance for Accelerating Excellence in Science in Africa (AESA) and supported by the New Partnership for Africa's Development Planning and Coordinating Agency (NEPAD Agency) with funding from the Wellcome Trust [grant # 107752/Z/15/Z] and the UK government. The views expressed in this publication are those of the author(s) and not necessarily those of AAS, NEPAD Agency, Wellcome Trust or the UK government. R.M.M was supported by TESA II which is part of the EDCTP2 programme supported by the European Union (grant number 1051-TESAII-RegNet2015). The views and opinions of authors expressed herein do not necessarily state or reflect those of EDCTP. S. M. was supported in part by the Fogarty International Center and National Institute of Mental Health of the National Institutes of Health under Award Number D43 TW010543. The funders had no role in study design, data collection and analysis, decision to publish, or preparation of the manuscript.

**Competing interests:** The authors have declared that no competing interests exist.

## Conclusions

The IH-Int assay had a high amplification success rate and high concordance with the commercial assay. It is significantly cheaper compared to the commercial assay. Our assay has the needed specifications for routine monitoring of participants on Dolutegravir based regimens in Botswana.

## Introduction

The global number of people living with HIV (PLWH) has substantially increased over the past two decades; from 16.9 million by the end of 1994 to 36.9 million PLWH by the end of 2017 [1, 2]. Global access to combination antiretroviral therapy (cART) has also increased resulting in a pronounced decline in morbidity and mortality amongst PLWH compared to those who are not on cART [2–4]. This trend is being seen and replicated globally especially with the global scale-up of universal access to cART [2, 5].

Over 20 antiretrovirals drugs (ARVs) spanning 7 drug classes are available in the armament of anti-HIV medicines available to PLWH [6]. They are available either as single tablet or a combination of two or more ARVs in single tablet regimens (STR) [6].

Dolutegravir (DTG) is a newer, potent, second generation integrase strand transfer inhibitor (INSTI) that is available as a STR combination with tenofovir disoproxil fumarate (TDF) and lamivudine (3TC) [TLD] at an affordable cost to low and middle income countries (LMICs) such as Botswana [7–10]. A majority of countries in sub-Saharan Africa (SSA), a region that is highly burdened by the HIV epidemic will soon have access to TLD [8].

However, there is a rise in pre-treatment and acquired HIV drug resistance (HIVDR) mutations to 'backbone' ARVs such TDF, 3TC and non-nucleoside reverse transcriptase inhibitors (NNRTIs) and this is more pronounced in SSA [11, 12]. This could increase the possibility of placing patients on a 'functional' DTG monotherapy, which can contribute to development of resistance to DTG. A recent modelling study projects that the impact of HIVDR on AIDS deaths, new infections and ART costs will increase in SSA between year 2016–2030 if pre-treatment HIVDR rates are over 10% [13]. The scale-up of TLD will increase the demand for INSTI drug resistance testing. This will be pronounced amongst PLWH previously exposed to first generation INSTIs such as raltegravir (RAL) as is the case in Botswana and other SSA countries.

DTG has a better safety profile, higher genetic barrier to resistance and better clinical efficacy compared to other ARVs such as efavirenz [9, 10, 14, 15]. However, PLWH with prior exposure to RAL that developed a 'Q148 mutant virus' might have reduced susceptibility to DTG reducing its effectiveness [16, 17]. This has also been demonstrated in case reports from resource limited settings (RLS) in-which individuals with prior RAL exposure who were commenced on DTG combination antiretroviral therapy (cART) developed treatment failure with the selection of the Q148 resistance pathway [18, 19].

Resource rich settings recommend HIV genotypic drug resistance testing (GRT) of the reverse transcriptase (RT) and protease (PR) regions prior to ART initiation and at other time points to guide optimal ARV selections [20, 21]. Due to the low rates of INSTI transmitted drug resistance (TDR) mutations, GRT at the integrase region is not routinely recommended unless clinically indicated such as when patients are experiencing virological failure (VF) while on INSTI cART or if there is a high risk of INSTI TDR mutations [20]. In contrast, the World Health Organisation recommends a surveillance and monitoring approach to tackling HIV

drug resistance [22]. INSTI TDR have been previously reported [23–25] as well as development of INSTI resistance associated mutations (RAMs) with subsequent virological failure amongst ARV treatment experienced individuals in RLS [18, 26].

There are limited options for GRT and most of these are traditionally prohibitively expensive and have been optimised for HIV-1 clade B strain which is not the predominant clade found in SSA. The ViroSeq™ HIV-1 Integrase RUO Genotyping kit (Celera Corporation, USA) (VS-Int) is a commercial assay that detects for HIVDR mutations in the integrase region of *pol* gene. Due to the high cost associated with HIVDR commercial assays, many laboratories have developed less expensive optimised in-house Sanger based [27] GRT assays ("home brews") specific for their most common circulating HIV strains [28–31].

Botswana, with a predominantly HIV-1C epidemic, introduced DTG into its ART treatment programme in June 2016 and currently relies on commercial assays to test for INSTI HIVDR. Genotypic HIVDR testing typically costs between 140–380 USD/ test hindering their use in resource limited settings (RLS) such as in Botswana [32, 33,34].

Increased access to newer classes of ARVs like DTG in resource limited settings (RLS) where often access to HIVDR testing is limited raises the possibility of an increase in INSTI RAM rates. There is an urgent need for more frequent VL monitoring and cost effective in-house INSTI HIVDR testing to ensure effectiveness of DTG based ART.

We therefore sought to optimise and evaluate an in-house "home-brew" integrase drug resistance assay (IH-Int) against a commercial assay, VS-Int, for routine use in Botswana and other countries with a background of HIV-1C infection.

## Materials and methods

### Study population, specimen panel and preparations

A total of fifty-five (55) stored plasma samples were used in our assessment. We randomly selected fifty four (54) -80˚C stored plasma samples from a previously completed study (BHP063) [35]. This study enrolled ART naïve HIV-1 recently infected individuals between January 2012 and December 2015 and it's described elsewhere [35]. The median plasma HIV-1 RNA load (viral loads [VL]) was 4.3 [Q1, Q3 (3.7, 4.9)] $\log_{10}$ copies/mL quantified by Abbott m2000sp/Abbott m2000rt (Wiesbaden, Germany).

One (1) well characterized plasma sample collected in 2017 from a highly treatment experienced patient who was enrolled in the Botswana Epidemiological ART Treatment Cohort Study (BEAT) and described elsewhere was included in the analysis as it had many INSTI RAMs determined before by the VS-Int assay [18]; The VL was 2.7 $\log_{10}$ copies/ml quantified by Cobas TaqMan/Cobas Ampliprep 48 and 96 systems (Roche Diagnostics, Branchburg, USA).

The Abbott and Cobas TaqMan/Cobas Ampliprep assays for VL quantification and HIV drug resistance testing were all performed at the Botswana Harvard HIV Reference Laboratory (BHHRL) in Gaborone, Botswana. BHHRL is a SADCAS ISO 15189 accredited laboratory and maintains certification of the virology assays through Rush University's virology quality assurance programme.

### Ethical considerations

Ethical clearance for BHP063 and BEAT studies was obtained from the Human Resource Development Committee in Gaborone, Botswana; protocol # HRDC 0638 and HPDME-13/ 18/1 XI (150) respectively. In addition, BHP063 protocol was approved by the human subjects committee of Harvard T.H Chan School of Public Health (protocol # 20770).

## RNA extractions, PCR amplifications and sequencing

**VS-Int assay.**   Genotyping using the VS-Int commercial assay was performed as per the manufacturer's instructions. Briefly, HIV-1 viral RNA was manually isolated from 500μL of participant's plasma by cold pelleting under high-speed centrifugation followed by cell lysis, isopropranolol precipitation and cold ethanol-based re-suspension with elution of 30 μL of HIV RNA. A subsequent one step RT-PCR reaction was performed utilising 10μL of the re-suspended viral RNA. The VS-Int sample preparation and genotyping kits contained all the reagents needed for extraction, reverse transcription polymerase chain reaction (RT-PCR) and sequencing steps that utilise four sequencing primers.

**IH-Int assay.**   Briefly, HIV-1 RNA was automatically extracted from 400μL of plasma by using EZ1 Virus Mini Kit v2.0 (Qiagen, Valencia, CA, USA) cartridges that were loaded on an automated EZ1 Advanced XL (Qiagen) machine. A one step RT-PCR reaction mix was prepared comprising; 0.5μL of Transcriptase Enzyme Roche One Step (Roche, Indianapolis, IN, USA), 7 μL of deionised water (dH$_2$0), 5 μL of Buffer 5X, 2.5 μL primer mix of 2 μM INFORI (5' GGA ATC ATT CAA GCA CAA CCA GA 3' nucleotide positions relative to HXB2 4059–4081) and INREV-1 (5'–TCT CCT GTA TGC AGA CCC CAA TAT–3' 3' nucleotide positions relative to HXB2 5244–5267) and 10 μL of the extracted viral RNA template for one reaction mix volume totalling 25 μL.

RT-PCR thermal cycling conditions were; one cycle of 50˚C for 30 mins, one cycle of 94 ˚C for 7 minutes, 10 cycles of 94 ˚C for 10 seconds, 52.5 ˚C for 30 seconds, 68 ˚C for 2 minutes, 35 cycles of 94 ˚C for 10 seconds, 53 ˚C for 30 seconds, 68 ˚C for 2 minutes that increased by 10 seconds with each additional cycle with a final extension step of 68 ˚C for 5 minutes and a hold at 4 ˚C. A PCR product of about 1300bp was generated that covered the complete HIV-1 integrase region of *pol*. For both assays, only one attempt to obtain PCR or sequencing result was used, no repeat testing of samples using both assays was allowed.

The primers and reaction mixes components were adapted and modified from experiments conducted on a population largely infected with a clade C virus [28]. The PCR thermal cycling conditions were adapted and modified from experiments described elsewhere [35, 36]. The RT-PCR products where run on 1% Agarose gel immersed in 1X TBE buffer stained with 3 μL of Ethidium bromide for about 30 mins at 100 volts to verify amplification and correct size of amplicons.

## Sequencing

**VS-Int assay.**   Four sequence mixes (forward primers A and B, reverse primers C and D) that came with the VS-Int genotyping kits were each combined with 8μL of the purified, diluted (where necessary) RT-PCR product. Cycle sequencing parameters (25 cycles) were 96 ˚C for 10 seconds, 50 ˚C for 5 seconds, 60 ˚C for 4 minutes and a hold at 4 ˚C.

Sanger sequencing for VS-Int and IH-Int was performed using an Applied Biosystems 3130xL Genetic Analyser (Applied Biosystems, California, CA, USA).

**IH-Int.**   Column purification of amplicons was performed with QIAquick PCR purification kits (Qiagen, Hilden, Germany). BigDye terminator cycle sequencing ready reaction kit version 3.1 (Applied Biosystems, Carlsbad CA, USA) was used for Sanger sequencing utilising four primers; HIV+4141 (5' TCT ACC TGG CAT GGG TAC CA 3' nucleotide positions relative to HXB2 4141–4160), INFORI (5' GGA ATC ATT CAA GCA CAA CCA GA 3' nucleotide positions relative to HXB2 4059–4081), INREVII (5' CCT AGT GGG ATG TGT ACT TCT GA 3' nucleotide positions relative to HXB2 5197–5219 and IN4764AS (5' CCATTTGTACTGCTGTCTTAA 3' nucleotide positions relative to HXB2 4764–4744).

The cycle sequencing reaction mix contained 3.8 μL of $dH_20$, 3 μL of Big Dye 5X sequencing buffer, 1 μL Big Dye terminator, 0.2 μL of 10 μM of each sequencing primer and 2 μL of the purified DNA template to make a total reaction volume of 10 μL. Cycle sequencing parameters were the same as for those used for the VS-Int assay.

Purification of cycle sequencing products was done using ZR-96 DNA sequencing clean-up kit (Zymo research, Irvine, CA, USA) according to manufactures instructions.

## Sequence, phylogenetic and mutational analysis

Electropherograms obtained were manually assembled and edited using Sequencher® version 5.0 DNA sequence analysis software (Gene Codes Corporation, Ann Arbor, MI, USA) [37] for both the IH-Int and VS-Int assays respectively. The assembly parameters used were a minimum match percentage of 85% and a minimum overlap of 20 base-pairs for sequences derived by both the IH-Int and VS-Int assays. The generated FASTA files were exported to BioEdit version 7.2.0 software for further analysis [38]. Multiple sequences were aligned using ClustalW Multiple alignment programme [39] embedded in BioEdit [38] using the HXB2 (accession number K03455.1) as reference. The 88 integrase sequences generated by the VS-Int and IH-Int methods were then compared for quality assurance and clustering using molecular phylogenetic analysis by maximum likelihood method in MEGA 7 software [40]. A Phylogenetic tree was constructed using Tamura-Nei substitution model with gamma distribution rates among sites and inferred from 1000 bootstrap replicates [41].

**Table 1. Baseline demographics and viral load characteristics of 55 HIV-1C infected individuals.**

| Characteristic | n (%) |
|---|---|
| Male | 11 (20) |
| Female | 44 (80) |
| Age in years Median, (Q1, Q3) | 29, (25, 34) |
| VL >1,000 cps/ml; median $\log_{10}$ VL (Q1, Q3) copies/ml | 48 (87.3); 4.5 (3.9,4.9) |
| VL <1,000 cps/ml; median $\log_{10}$ VL (Q1, Q3) copies/ml | 7 (12.7); 2.7 (2.2,2.7) |

All the study participants were from Botswana. VL, viral load; cps/ml, copies/mL

**Table 2. RT-PCR amplification and sequencing success rates of IH-Int and VS-Int assays with further stratification according to viral loads.**

| | | IH-Int | VS-Int |
|---|---|---|---|
| Amplification status of participant samples (n, %) n = 55; median $\log_{10}$ VL (Q1, Q3) cps/ml | Yes; | 53 (96.4); 4.4 (3.7, 4.8) | 45(81.8); 4.5 (3.9, 4.8) |
| | No, | 2 (3.6); 2.1 & 2.7 $\log_{10}$cps/ml | 10 (18.2); 3.5 (2.8, 4.5) |
| Amplification success rate of samples with VL>1,000 cps/ml, n = 48, (%); median $\log_{10}$ VL (Q1, Q3) cps/ml | | 48 (100); 4.5 (3.9, 4.9) | 41 (85.4); 4.5 (4.1, 4.9) |
| Amplification success rate of samples with VL<1,000 cps/ml, n = 7, (%); median $\log_{10}$ VL (Q1, Q3) cps/ml | | 5, (71.4); 2.7 (2.6, 2.7) | 4, (57.1); 2.7 (2.7, 2.8) |
| Sequencing success rate | | 50/55 (90.9) | 38/55(69.1) |

The table shows the two assays RT-PCR performance stratified according to viral loads (VL) less than and greater than 1,000 copies/m. VL, viral load; cps/ml, copies/mL; IH-Int, in-house integrase drug resistance assay; VS-Int, ViroSeq™ HIV-1 Integrase sample preparation and Genotyping kit (Celera Corporation, USA); RT-PCR, reverse transcribed polymerase chain reaction.

Mutational analysis of sequences was performed using the Stanford HIV drug resistance database algorithm version 8.7 (https://hivdb.stanford.edu/hivdb/by-sequences/) [42]. HIV-1 subtype was determined using REGA HIV-1 subtyping tool version 3.0 (http://dbpartners. stanford.edu:8080/RegaSubtyping/stanford-hiv/typingtool/) [43].

### Accession numbers

The 33 pairs of nucleotide sequences obtained in our study were submitted to national center for biotechnology information (NCBI) GenBank and their accession numbers are MN037428 to MN037493. Additional nucleotide sequences from our study are available in Genbank under accession numbers MN462669 to MN462690.

### Cost comparison

We calculated the reagents costs by analysing a batch of 12 samples used for the IH-Int and VS-Int assay. Estimated costs involved in the IH-Int and VS-Int assay included all stages from extraction, RT-PCR and sequencing.

### Statistical analysis

Statistical analysis was performed using STATA version 14 (Stata Corp, College Station, TX, USA. Major and minor HIVDR mutations obtained from both assays were compared as obtained from the Stanford HIV drug resistance database.

## Results

### Comparison of PCR and sequencing success rates between the VS-Int and IH-Int assay

From the 55 samples, 44 (80%) and 11 (20%) were from female and male participants respectively. Other baseline characteristics are shown in Table 1. 45 (81.8%) of the samples amplified with the VS-Int assay and 53 (96.4%) amplified with the IH-Int assay, Table 2. Of the 10/55

**Table 3. Samples that failed amplification with each assay, VS-Int (table A) and IH-Int (table B) and how they faired when run with a different assay, IH-Int (table A) and VS-Int (table B) and associated viral loads.**

| A | PID | VL $log_{10}$ copies/mL | Amplifica-tion with VS-Int | Amplifica-tion with IH-Int | B | PID | VL $log_{10}$ copies/mL | Amplifica-tion with IH-Int | Amplifica-tion with VS-Int |
|---|-----|----|----|----|---|-----|----|----|----|
| 1 | KKSG41 | 2.1 | N | N | | KKSG41 | 2.1 | N | N |
| 2 | KKSG42 | 2.2 | N | Y | | KKSG51 | 2.7 | N | Y |
| 3 | KKSG43 | 2.7 | N | Y | | | | | |
| 4 | KKSG44 | 3.2 | N | Y | | | | | |
| 5 | KKSG45 | 3.4 | N | Y | | | | | |
| 6 | KKSG46 | 3.5 | N | Y | | | | | |
| 7 | KKSG47 | 3.7 | N | Y | | | | | |
| 8 | KKSG48 | 4.8 | N | Y | | | | | |
| 9 | KKSG49 | 5.4 | N | Y | | | | | |
| 10 | KKSG50 | 5.7 | N | Y | | | | | |

Table 3A and B shows the amplification successes of samples that failed amplification with either the VS-Int or IH-Int assay when they were tested with a different assay. PID, participant identification number; N, No; Y, Yes; VL, viral load; IH-Int, in-house integrase drug resistance assay; VS-Int, ViroSeq™ HIV-1 Integrase sample preparation and Genotyping kit (Celera Corporation, USA).

(18.2%) samples that failed amplification with the VS-Int assay, 9/10 (90%) were able to be amplified by the IH-Int assay, Table 3.

The VS-Int and IH-Int were both a one-step RT-PCR and utilised 4 sequencing primers that covered all of integrase region. Both assays had a similar hands-on and protocol required times, (Fig 1). The VS-Int and IH-Int assays had a sequencing success rate of 38/45 (84.4%) and 50/53 (94.3%), respectively, (Fig 2).

Further analysis of both assays on samples with VL greater than 1,000 copies/ml revealed an amplification success rate of 41/48 (85.4%) for the VS-Int and 48/48 (100%) for the IH-Int assay, Table 2. A total of three INSTI mutations were identified by the IH-Int assay and two INSTI mutations were identified by the VS-Int assay, Table 4.

From the 50 sequences obtained by the IH-Int assay and 38 sequences obtained by the VS-Int assay, 33 paired sequences were identified. The low number of paired sequences was due to failures in amplification and sequencing by the VS-Int assay. A total of 33 paired

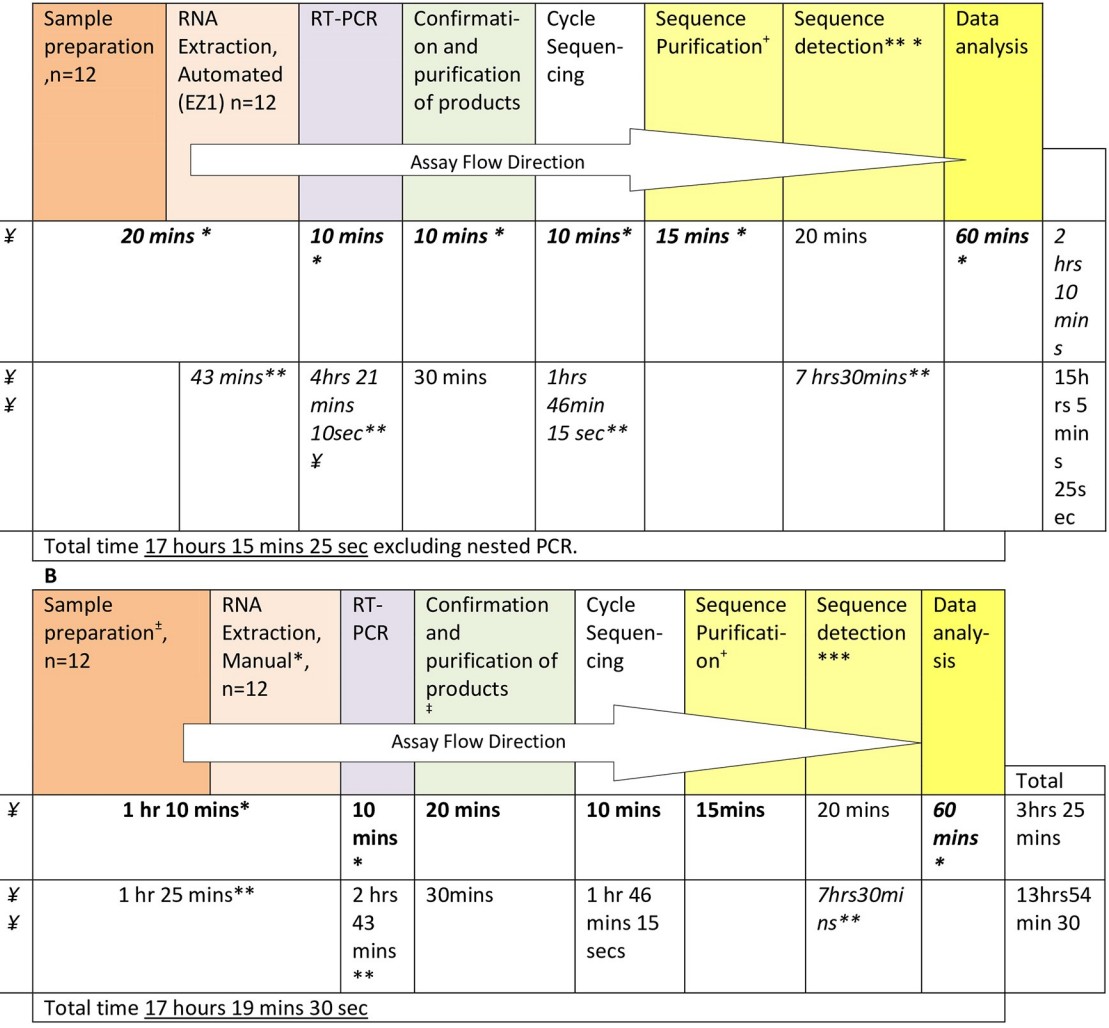

**Fig 1. HIV drug resistance testing workflow using an IH-Int and VS-Int assay (panel B).** *estimated hands-on time, ** fixed machinery times, #Thermocycler times, + using Zymogen purification kit;*** on 8 capillary ABI PRISM 3130 xl Genetic Analyser for 12 samples, ‡ViroSeq HIV-1 Integrase Sample Prep Kit 4J94-72, ‡ purification (Exonuclease 1). RT-PCR, reverse transcriptase-polymerase chain reaction; IH-Int, in-house "home-brew" integrase drug resistance assay; VS-Int, ViroSeq™ HIV-1 Integrase sample preparation and Genotyping kit (Celera Corporation, USA).

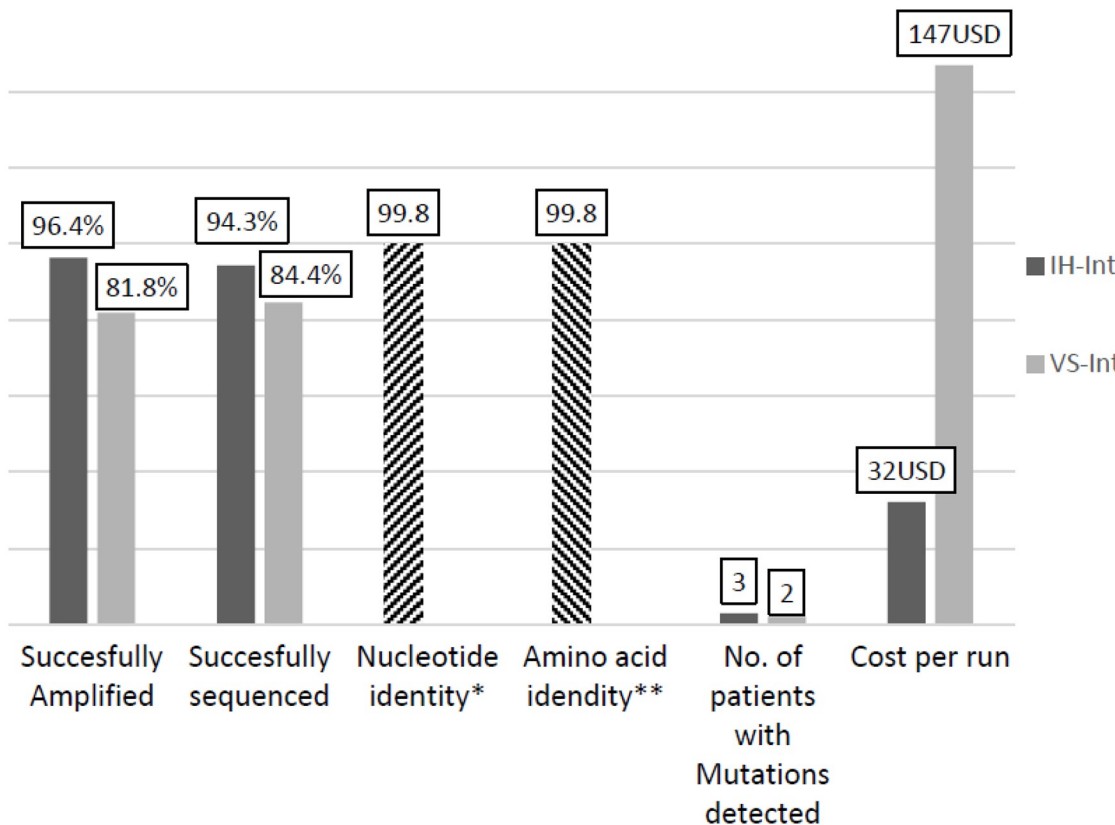

**Fig 2. Differences between amplification and sequencing success rates and cost per test run of two assays.** Viral loads for samples successfully amplified by IH-Int, mean (Q1, Q3); 4.4 (3.7, 4.8), 2.2 copies/ml. Viral loads for samples successfully amplified by VS-Int, mean (Q1,Q3),minimum; 4.4 (3.9, 4.8), 2.6 copies/ml. *S.D = ± 0.30; **S.D = ± 0.39. IH-Int, in-house integrase drug resistance assay; VS-Int, ViroSeq™ HIV-1 Integrase sample preparation and Genotyping kit (Celera Corporation, USA).

sequences revealed a mean amino acid and nucleotide similarity of 99.8% (SD = 0.39) and 99.8% (SD = 0.30) amongst the two assays. All sequenced samples were shown to be of a clade C virus by REGA HIV-1 and 2 subtyping tool [43].

The cost of IH-Int were estimated at 32 USD per sample and the VS-Int estimated at 147 USD to run one sample (reagent costs only) and both have similar run times Table 5, Figs 1–3.

**Table 4. Comparison of HIV-1 Integrase drug resistance mutations from three patients detected by sanger sequencing using the IH-Int and VS-Int assay and associated viral loads.**

| PID | Source | Viral Load cps/ml | Genotyped region | IH-Int detected mutations | VS-Int detected mutations |
|---|---|---|---|---|---|
| | | | | Integrase region positions | Integrase region positions |
| KKSG19 | Plasma | 93,154 | Integrase | T97A | T97TA |
| P1 | Plasma | 4,745 | Integrase | E157Q | N/A* |
| KKSG06 | Plasma | 515 | Integrase | E138K, G140A, S147G, Q148R, T97A | E138K, G140A, S147G, Q148R, T97A |

PID, participant identification number; IH-Int, in-house integrase drug resistance assay; VS-Int, ViroSeq™ HIV-1 Integrase sample preparation and Genotyping kit (Celera Corporation, USA); cps/ml, copies/ml;

* failed to amplify.

**Table 5. Comparison between IH-Int assay and commercial VS-Int resistance assays.**

|  | **IH-Int assay** | **VS-Int assay** |
|---|---|---|
| Description of assay | One step using 4 sequencing primers Has option of utilising a nested RT-PCR step but this wasn't used in our comparison | One step RT-PCR using 4 sequencing primers No option of performing a nested PCR step |
| Region targeted and positions covered | HIV-1 Integrase region from codons 1–288 | HIV-1 Integrase region from codons 1–288 |
| Input sample type | Plasma | Plasma |
| Volume (µL) | 200–400 | 500 |
| Sample preparation and RNA extraction | Automated EZ1 or manual Qiagen | Manual |
| Sequencing type | Sanger sequencing using Big Dye chemistry on an ABI PRISM capillary based system | Sanger sequencing using Big Dye chemistry on an ABI PRISM capillary based system |
| Time to results | 17–18 hrs | 17–18 hrs |
| Estimated Costs[+] | 32 USD | 147 USD [*+] |
| Technical skills | High | High |
| Laboratory set-up | BSL Level 2 or above | BSL Level 2 or above |
| Weaknesses | Requires costly infrastructure, high technical skills Not yet commercially available | Requires costly infrastructure, high technical skills, prohibitively expensive |
| Strength's | Very affordable | Commercially available, closed, standardized system from extraction, PCR, sequencing and sequence analysis |

[+]reagent cost only, [*]For comparison, HIV-1 PR and RT by ViroSeq® assay is about 155–380 USD cost/test (excluding labour) [33], [+] HIV DNA GRT RT is 286 USD and GRT DNA/RNA is 143 USD [32].

This table is modified and adapted from table 1, supplementary data of [33] and table 5 of [34].

IH-Int, in-house integrase drug resistance assay; VS-Int, ViroSeq™ HIV-1 Integrase sample preparation and Genotyping kit (Celera Corporation, USA); GRT, genotypic resistance test; RT-PCR, reverse transcribed polymerase chain reaction; USD, united states dollars; VL, viral load; PR, protease; RT, reverse transcriptase; BSL, Biosafety level.

## Discussion

Scale-up of dolutegravir based regimens (DBR) in resource limited settings where HIV-1 clade C is prevalent warrants an urgent need for affordable genotyping HIVDR testing technologies for monitoring INSTI RAMs for patient care and drug resistance surveillance [8, 44]. In this study, we optimised an in-house HIV integrase drug resistance assay and compared it against a commercial ViroSeq™ HIV-1 Integrase sample preparation and genotyping assay for routine monitoring of drug resistance in Botswana.

Our in-house integrase drug resistance assay was able to amplify 100% (n = 48) of patient plasma samples with viral loads greater than 1,000 copies/ml compared to 85.4% (n = 48) amplification success rate with the ViroSeq™ HIV-1 Integrase sample preparation and genotyping assay. Similar VS-Int amplification success rates have also been reported by others [15, 30].

In addition, our IH-Int assay was able to amplify 9/10 (90%) of samples that failed amplification with the ViroSeq™ HIV-1 integrase genotyping assay. The IH-Int assay was able to perform this at approximately a quarter of the cost of the commercial assay.

The higher success rate of the IH-Int could be due to the IH-Int assay having primers that are specific for HIV-1C, while the VS-Int assay primers were designed to cover various HIV-1

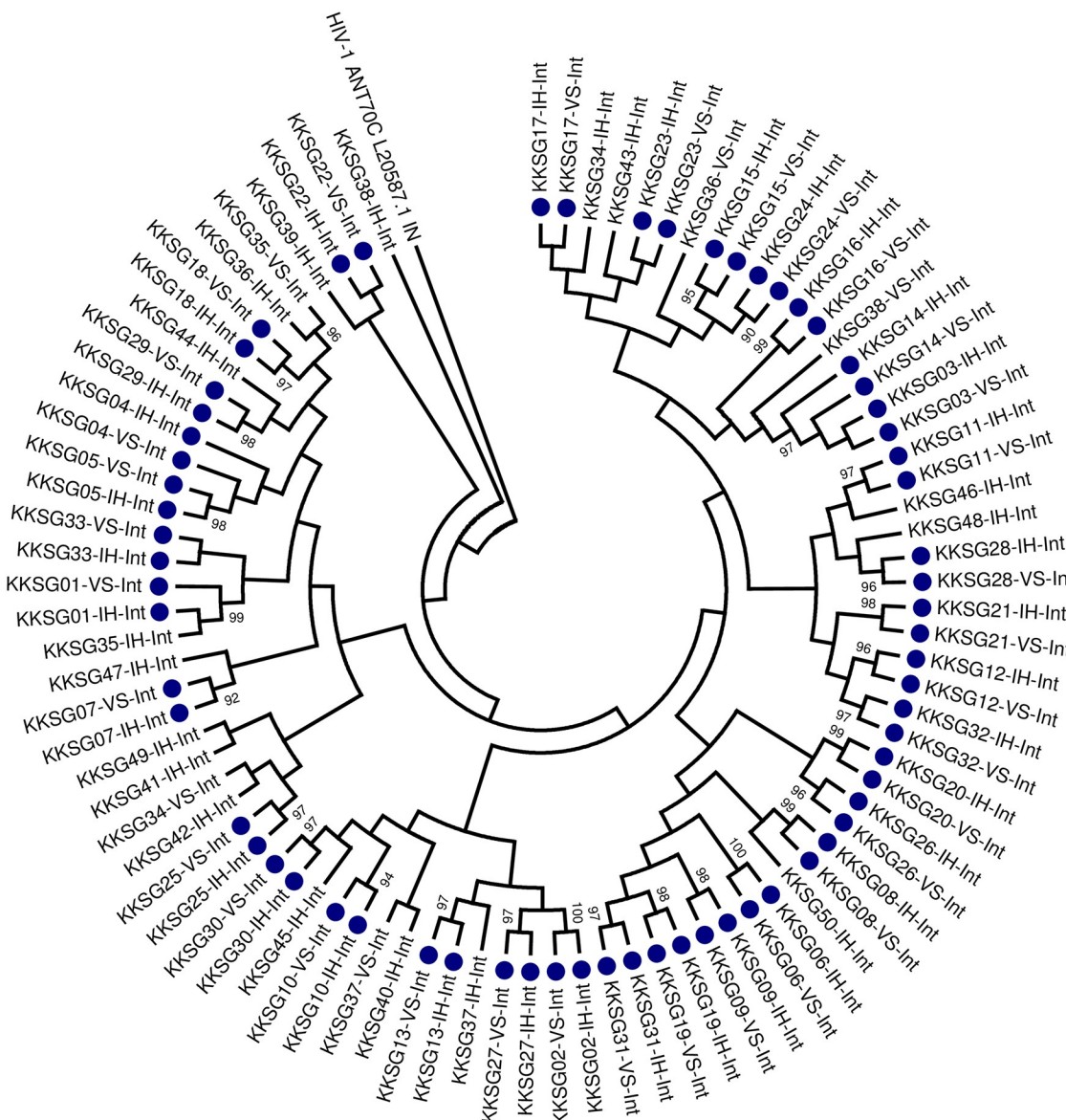

**Fig 3. Molecular phylogenetic analysis by maximum likelihood method of 88 integrase sequences derived by the ViroSeq™ HIV-1 integrase RUO genotyping kit (VS-Int) and in-house "home-brew" integrase drug resistance assay (IH-Int).** The numbers next to the nodes represent Bootstrap values (1000 replicates) >90.

subtypes. There is limited published data on the performance of the ViroSeq™ HIV-1 Integrase sample preparation and genotyping assay in HIV-1 non subtype B strains.

Previous studies have shown that the ViroSeq™ Protease-RT assay is less successful with non B HIV-1 subtypes [45, 46]. It is most likely that the VS-Int assay has the same limitations as the ViroSeq™ Protease-RT assay.

At 99.8% and 99.8% nucleotide and amino acid identity respectively between the IH-Int and the VS-Int assays, this demonstrates that the sequences generated by the two assays are similar. The results of the sequences of the patient with multiple INSTI resistance mutations attest to this good concordance with all the INSTI resistance mutations identified by the VS-Int assay being identified with the IH-Int assay.

The profound cost differences associated with our IH-Int assay in the era of DBR scale-up are encouraging. This high level of RT-PCR success rates and cost savings is similar to what has been traditionally observed with other in-house HIV drug resistance assays [30, 31, 47, 48].

We did not evaluate our assay amongst non-clade C panel of HIV viruses, as most countries in sub-Saharan Africa have a predominantly clade-C epidemic which may makes our assay applicable to these settings. Our sample size and volumes did not permit an evaluation of reproducibility and specificity hence they were not assessed but are in the future directions. A better in-house HIVDR assay for patient care would be one that quantifies viral loads, measures ARV drug levels and detects for drug resistance mutations in the integrase, reverse transcriptase and protease region of HIV-1 *pol* gene in one step which could ideally be packaged into a point of care test.

In conclusion, our in-house assay had a high amplification success rate and high concordance with the commercial assay and it is significantly less expensive than the commercial assay.

Our In-house integrase assay has the required specifications to be used in HIV-1 clade C infected individuals for routine monitoring of integrase RAMs in Botswana.

## Acknowledgments

We thank our patients and staff of Princess Marina Hospital Infectious Disease Care Clinic (PMH IDCC). We are grateful to the Botswana Harvard HIV Reference Laboratory staff, Botswana Harvard Partnership and Botswana Ministry of Health and Wellness for their collaboration. We also thank Thongbotho Mphoyakgosi, Tshenolo Ntsipe and Segomotso Maphorisa for their outstanding laboratory assistance. We acknowledge Tefo Zongwa and Itseng Bame Montle for their input on the costing analysis. Finally we thank Doreen Ditshwanelo and Naledi T. Gonnye for excellent editorial assistance.

## Author Contributions

**Conceptualization:** Kaelo K. Seatla, Simani Gaseitsiwe.

**Data curation:** Kaelo K. Seatla, Dorcas Maruapula, Christopher F. Rowley, Ava Avalos, Simani Gaseitsiwe.

**Formal analysis:** Kaelo K. Seatla, Wonderful T. Choga, Mompati Mogwele, Thabo Diphoko, Lucy Mupfumi, Ishmael Kasvosve, Sikhulile Moyo, Simani Gaseitsiwe.

**Funding acquisition:** Simani Gaseitsiwe.

**Investigation:** Kaelo K. Seatla, Wonderful T. Choga, Mompati Mogwele, Thabo Diphoko, Dorcas Maruapula.

**Methodology:** Kaelo K. Seatla.

**Project administration:** Kaelo K. Seatla, Rosemary M. Musonda, Christopher F. Rowley, Ishmael Kasvosve, Sikhulile Moyo, Simani Gaseitsiwe.

**Resources:** Christopher F. Rowley, Simani Gaseitsiwe.

**Supervision:** Rosemary M. Musonda, Christopher F. Rowley, Ava Avalos, Ishmael Kasvosve.

**Validation:** Kaelo K. Seatla.

**Visualization:** Kaelo K. Seatla.

**Writing – original draft:** Kaelo K. Seatla.

**Writing – review & editing:** Kaelo K. Seatla, Wonderful T. Choga, Mompati Mogwele, Thabo Diphoko, Dorcas Maruapula, Lucy Mupfumi, Rosemary M. Musonda, Christopher F. Rowley, Ava Avalos, Ishmael Kasvosve, Sikhulile Moyo, Simani Gaseitsiwe.

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
