## [Decision Letter · Decision Letter 0]

23 Jul 2019

PONE-D-19-15741

Validation of an affordable and sensitive in-house HIV-1 subtype C integrase genotyping assay

PLOS ONE

Dear 65Dr Gaseitsiwe,

Thank you for submitting your manuscript to PLOS ONE. After careful consideration, we feel that it has merit but does not fully meet PLOS ONE’s publication criteria as it currently stands. Therefore, we invite you to submit a revised version of the manuscript that addresses the points raised during the review process.

There are some very helpful comments from the two reviewers. I would encourage you to address these in your revision. Please note there are some additional comments from reviewer #2 in the attached file - let me know if you are unable to access the file/comments.

We would appreciate receiving your revised manuscript by Sep 06 2019 11:59PM. To enhance the reproducibility of your results, we recommend that if applicable you deposit your laboratory protocols in protocols.io, where a protocol can be assigned its own identifier (DOI) such that it can be cited independently in the future. For instructions see: http://journals.plos.org/plosone/s/submission-guidelines#loc-laboratory-protocols

We look forward to receiving your revised manuscript.

Kind regards,

Richard John Lessells, BSc, MBChB, MRCP, DTM&H, DipHIVMed, PhD

Academic Editor

PLOS ONE

Journal Requirements:

2) We note that you have stated that you will provide repository information for your data at acceptance. Should your manuscript be accepted for publication, we will hold it until you provide the relevant accession numbers or DOIs necessary to access your data. If you wish to make changes to your Data Availability statement, please describe these changes in your cover letter and we will update your Data Availability statement to reflect the information you provide.

Reviewers' comments:

Reviewer's Responses to Questions

**Comments to the Author**

1. Is the manuscript technically sound, and do the data support the conclusions?

Reviewer #1: Yes

Reviewer #2: Partly

2. Has the statistical analysis been performed appropriately and rigorously? 

Reviewer #1: N/A

Reviewer #2: Yes

3. Have the authors made all data underlying the findings in their manuscript fully available?

Reviewer #1: Yes

Reviewer #2: No

4. Is the manuscript presented in an intelligible fashion and written in standard English?

Reviewer #1: Yes

Reviewer #2: Yes

5. Review Comments to the Author

Reviewer #1: PONE-D-19-15741

Validation of an affordable and sensitive in-house HIV-1 subtype C integrase genotyping assay

This manuscript describes the technical validation of an HIV-1 integrase in-house genotyping assay and compares the cost with its commercial counterpart. Given the recent introduction of integrase inhibitors in many countries, this is a valuable contribution to the scientific community, however there are some limitations to the work.

Major comments:

1. When processing the samples, were repeat testing for PCR and/or sequencing allowed or was only 1 attempt to obtain a result allowed. This should be included in the materials and methods. Especially the poor sequencing success rate is unexpected for samples with a suitable PCR product

2. At what level was mixed based calling performed? 20%, 25%, were the same rule applied for VS-Int and IH-Int since different sequence analysis software was used.

3. Why were there only 33 paired sequences available, is this caused by the inability to obtain good sequence quality for a number of specimens using VS-Int

4. Table 4, I don’t think it is correct to say that VS-Int did not pick up the mutation (nil) because there was no PCR product available for this specimen. I would rather change nil to N/A or even remove it from the table

5. I think the statement on line 359 regarding the IH assay’s ability to detect more mutations is incorrect. If sample P1 failed to amplify using VS, one cannot make a statement regarding the inability to detect that mutation. For KKSG19 the mutation T97A was detected by both assays, albeit it being a mix for VS. In addition, based on the limited number of specimens with DRMs, no strong statement should be made regarding the ability to detect mutations.

6. Clarify statement on line 371-372: “there is also a lower… by the VS Int assay.

7. Table 5 includes the option of utilising a nested RT-PCR step for IH-Int. Was nested PCR performed for any of the samples, if yes, for how many? And how did this impact the success rate of the IH assay?

8. Were reproducibility and specificity assessed for the IH assay. I think it is important to show that the same results (for PCR and sequencing) can be obtained when the same sample is repeated. If not yet done, reproducibility should be assessed and results should be included in the manuscript.

9. I understand that samples with InSTI mutations are difficult to find, but there have been reports from more than 1 patient from Botswana with InSTI mutations, are those samples available? If not, do you have access to EQA material with InSTI mutations? The inclusion of more samples with InSTI mutations would significantly improve the value of the manuscript

Minor comments:

1. Line 41-42: “…and for IH-int and VS-Int assay respectively” seems to be out of place in this sentence

2. Table 1: second last line looks like an error

3. Table 2 on page 13 seems to be a repeat from table 2 on page 10

4. I don’t see the value of figure 1 as primers seem to cover the same target of interest (both aa 1-288), however the position of the primers seems to be different between the assays.

5. Figure 2 is illegible

Reviewer #2: Thank you for the opportunity to review this very important manuscript.

Major comments:

1. The validation appears to be incomplete as a number of experiments which are usually part of a start validation exercise are missing. The following experiments would have added more value to the validation:

i) Validation should ideally include a set of sample on know genotypes - preferably EQA samples or samples genotyped by another method other than the methods being compared.

ii) Serially diluting a high viral load sample 3-5 samples (each dilution having at least 3 replicates) to determine the dynamic range of amplification of the method being validated (and that of the comparator) - this is ideal done using EQA samples/panel of samples.

iii). Testing the replicates on different days or in different runs (The report does not indicated over how many runs and days the experiments were done).

Otherwise, this is just a head to head comparison of two methods.

2. The performance of the two assays should be measure with the percentage of genotypes obtained from the initial samples tested. The percentage of samples amplified is an interim result. So the main results in terms of the performances of the two assays should be 50/55 (91%) vs 38/55 (69%).

Minor comments

1. Since the primers and the method used for the in-house method were adopted and modified from a previously published paper, the authors should provide details of the new primers including sequences and HXB2 position.

2. The phylogenetic tree, figure 4, should contain all 88 sequences generated from sequencing using the two methods (50 from the In-house method and 38 from VS.

3. Figure 3 is not clear

4.All the sequences above should also be submitted to Genbank instead of just the 33 pairs.

5. Please the comments on the attached file for the rest of the comments.

6. PLOS authors have the option to publish the peer review history of their article (what does this mean?). If published, this will include your full peer review and any attached files.

Reviewer #1: No

Reviewer #2: Yes: Justen Manasa

---

## [Author Response · Author response to Decision Letter 0]

9 Sep 2019

Reviewer #1: PONE-D-19-15741

Validation of an affordable and sensitive in-house HIV-1 subtype C integrase genotyping assay

This manuscript describes the technical validation of an HIV-1 integrase in-house genotyping assay and compares the cost with its commercial counterpart. Given the recent introduction of integrase inhibitors in many countries, this is a valuable contribution to the scientific community; however there are some limitations to the work.

Major comments:

1. When processing the samples, were repeat testing for PCR and/or sequencing allowed or was only 1 attempt to obtain a result allowed. This should be included in the materials and methods. Especially the poor sequencing success rate is unexpected for samples with a suitable PCR product

Response: 

Thank you for pointing this out. Only one attempt to obtain results was allowed, there was no multiple testing on the same sample. We have included in the material and methods section, line 182 the following; “For both assays, only one attempt to obtain PCR or sequencing result was used, no repeat testing of samples using both assays were allowed.”

2. At what level was mixed based calling performed? 20%, 25%, were the same rule applied for VS-Int and IH-Int since different sequence analysis software was used.

Response: 

Thank you for pointing this out. We had attempted to keep the VS-INT manufacturer’s instruction. In the revised manuscript, in order to avoid bias we have reanalysed sequences using Sequencher 5.0 for both assays-the VS-Int and IH int. This re-analysis did not have any impact in the results. We have modified and included in the materials and methods section, under sub-heading Sequence, phylogenetic and mutational analysis from line 215 the following; “Electropherograms obtained were manually assembled and edited using Sequencher® version 5.0 DNA sequence analysis software (Gene Codes Corporation, Ann Arbor, MI, USA) (36) for both the IH-Int and VS-Int assays respectively. The Sequencher assembly parameters used were a minimum match percentage of 85% and a minimum overlap of 20 base pairs for sequences derived by both the IH-Int and VS-Int assays”.

3. Why were there only 33 paired sequences available, is this caused by the inability to obtain good sequence quality for a number of specimens using VS-Int

Response: 

We have included the following in line 307; “From the 50 sequences obtained by the IH-Int assay and 38 sequences obtained by the VS-Int assay, 33 paired sequences were identified. The low number of paired sequences was due to failures in amplification and sequencing by the VS-Int assay”. 

As shown in Table 2 the amplification success rate for VS-Int was 82% as compared to 96% for the IH-Int, and for samples with viral load > 1000 copies/mL, the amplification success rate for VS-Int was 85% and 100% for the IH-Int assay.

4. Table 4, I don’t think it is correct to say that VS-Int did not pick up the mutation (nil) because there was no PCR product available for this specimen. I would rather change nil to N/A or even remove it from the table

Response: 

Thank you for pointing this out, in line 305 table 4 row 3, we have changed ‘nil’ to “N/A”.

5. I think the statement on line 359 regarding the IH assay’s ability to detect more mutations is incorrect. If sample P1 failed to amplify using VS, one cannot make a statement regarding the inability to detect that mutation. For KKSG19 the mutation T97A was detected by both assays, albeit it being a mix for VS. In addition, based on the limited number of specimens with DRMs, no strong statement should be made regarding the ability to detect mutations.

Response: 

Thank you for the comment. In line 353, we have modified the first sentence to reflect this clearly, we have deleted “…and was able to detect more INSTI RAMs…”. The sentence in line 353 now reads; “In addition, our IH-Int assay was able to amplify 9/10 (90%) of samples that failed amplification with the ViroSeq™ HIV-1 integrase genotyping assay”.

6. Clarify statement on line 371-372: “there is also a lower… by the VS Int assay.

Response: 

In line 365, we have deleted the sentence “There is also a lower likelihood that a mutation could be missed by the IH-Int assay that would otherwise be identified by the VS-Int assay”.

7. Table 5 includes the option of utilising a nested RT-PCR step for IH-Int. Was nested PCR performed for any of the samples, if yes, for how many? And how did this impact the success rate of the IH assay?

Response: 

Thank you for highlighting this point. Table 5, line 316, second column under heading “IH-Int assay”, we have modified as follows; “Has option of utilising a nested RT-PCR step available but this wasn’t used in our comparison”.

8. Were reproducibility and specificity assessed for the IH assay. I think it is important to show that the same results (for PCR and sequencing) can be obtained when the same sample is repeated. If not yet done, reproducibility should be assessed and results should be included in the manuscript.

Response: 

The tittle of our manuscript has been changed from “Validation of an affordable and sensitive in-house HIV-1 subtype C integrase genotyping assay” to “Comparison of an in-house ‘home-brew’ and commercial ViroSeq integrase genotyping assays on HIV-1 subtype C samples”.

In addition, we have added into the limitations under the discussion line 374 the following; “Our sample size and volumes did not permit an evaluation of reproducibility and specificity hence they were not assessed but are in the future directions.”

9. I understand that samples with InSTI mutations are difficult to find, but there have been reports from more than 1 patient from Botswana with InSTI mutations, are those samples available? If not, do you have access to EQA material with InSTI mutations? The inclusion of more samples with InSTI mutations would significantly improve the value of the manuscript

Response: 

Yes the reports of patients with InSTI mutations are correct. There are a few numbers of patients with InSTI mutations in Botswana but were not all part of this sample at the time of this study. We have 1 case of multi-class HIV DR mutations already included in this study. We plan further evaluations beyond subtype C and your suggestions are appreciated. We have since modified the title to reflect this as a comparison study as opposed to a validation.

Minor comments:

1. Line 41-42: “…and for IH-int and VS-Int assay respectively” seems to be out of place in this sentence

Response: 

We have modified the sentence and deleted the “…and…”. Line 45 now reads thus; “The mean nucleotide and amino acid similarity from 33 paired sequences was 99.8% (SD ± 0.30) and 99.8 % (SD ± 0.39) for the IH-Int and VS-Int assay respectively.”

2. Table 1: second last line looks like an error

Response: 

In table 1,line 263, we have deleted the second last line.

3. Table 2 on page 13 seems to be a repeat from table 2 on page 10

Response: 

This is correct, we have deleted table 2 on page 13.

4. I don’t see the value of figure 1 as primers seem to cover the same target of interest (both aa 1-288), however the position of the primers seems to be different between the assays.

Response: 

We have removed fig 1 from the manuscript

5. Figure 2 is illegible

Response: 

We have included the same fig 2 but with better resolution to improve illegibility. Figure 2 has now been renamed to figure1. In the former fig 2,now fig 1, we have deleted “Including nested PCR total time 21 hours 46 mins 35 sec”in panel A and “Total (excluding nested step)” caption in panel last right column.

Reviewer #2: Thank you for the opportunity to review this very important manuscript.

Major comments:

1. The validation appears to be incomplete as a number of experiments which are usually part of a start validation exercise are missing. The following experiments would have added more value to the validation:

i) Validation should ideally include a set of sample on know genotypes - preferably EQA samples or samples genotyped by another method other than the methods being compared.

Response: 

Thank you for pointing this out. The tittle of our manuscript has been changed from “Validation of an affordable and sensitive in-house HIV-1 subtype C integrase genotyping assay” to “Comparison of an in-house ‘home-brew’ and commercial ViroSeq integrase genotyping assays on HIV-1 subtype C samples”.

In addition, we have added into the limitations under the discussion line 374 the following; “Our sample size and volumes did not permit an evaluation of reproducibility and specificity hence they were not assessed but are in the future directions.”

ii) Serially diluting a high viral load sample 3-5 samples (each dilution having at least 3 replicates) to determine the dynamic range of amplification of the method being validated (and that of the comparator) - this is ideal done using EQA samples/panel of samples.

Response: 

The tittle of our manuscript has been changed from “Validation of an affordable and sensitive in-house HIV-1 subtype C integrase genotyping assay” to “Comparison of an in-house ‘home-brew’ and commercial ViroSeq integrase genotyping assays on HIV-1 subtype C samples”.

In addition, we have added into the limitations under the discussion line 374 the following; “Our sample size and volumes did not permit an evaluation of reproducibility and specificity hence they were not assessed but are in the future directions.”

iii). Testing the replicates on different days or in different runs (The report does not indicated over how many runs and days the experiments were done).

Otherwise, this is just a head to head comparison of two methods.

Response: 

Thank you for highlighting this. This is correct, indeed the tittle of our manuscript has been changed from “Validation of an affordable and sensitive in-house HIV-1 subtype C integrase genotyping assay” to “Comparison of an in-house ‘home-brew’ and commercial ViroSeq integrase genotyping assays on HIV-1 subtype C samples”.

We have added into the limitations under the discussion section about our sample size and volume of samples that did not permit an evaluation of reproducibility and specificity were not assed but are in the future directions. 

2. The performance of the two assays should be measure with the percentage of genotypes obtained from the initial samples tested. The percentage of samples amplified is an interim result. So the main results in terms of the performances of the two assays should be 50/55 (91%) vs 38/55 (69%).

Response: 

Thank you for pointing this out. This has been corrected in Table 2 line 264. The sequencing success rate has been changed to 50/55 (90.9) for the IH-Int and 38/55 (69.1) for the VS-Int

Minor comments

1. Since the primers and the method used for the in-house method were adopted and modified from a previously published paper, the authors should provide details of the new primers including sequences and HXB2 position.

Response: 

Under materials and methods, sub heading RNA extractions, PCR amplifications and sequencing, from line 170, we have modified to include the following; “A one step RT-PCR reaction mix was prepared comprising; 0.5µL of Transcriptase Enzyme Roche One Step (Roche, Indianapolis, IN, USA), 7 µL of deionised water (dH20), 5 µL of Buffer 5X, 2.5 µL primer mix of 2 µM INFORI (5' GGA ATC ATT CAA GCA CAA CCA GA 3' nucleotide positions relative to HXB2 4059-4081) and INREV-1 ( 5’-TCT CCT GTA TGC AGA CCC CAA TAT-3’ 3' nucleotides positions relative to HXB2 5244-5267) and 10 µL of the extracted viral RNA template for one reaction mix volume totalling 25 µL.”

In addition, the sequencing primers used for the IH-Int assay have been included in line 199 and reads thus; “IH-Int. Column purification of amplicons was performed with QIAquick PCR purification kits (Qiagen, Hilden, Germany). BigDye terminator cycle sequencing ready reaction kit version 3.1 (Applied Biosystems, Carlsbad CA, USA) was used for Sanger sequencing utilising four primers; HIV+4141 ( 5' TCT ACC TGG CAT GGG TAC CA 3' nucleotide positions relative to HXB2 4141-4160), INFORI ( 5' GGA ATC ATT CAA GCA CAA CCA GA 3' nucleotide positions relative to HXB2 4059 -4081), INREVII (5' CCT AGT GGG ATG TGT ACT TCT GA 3' nucleotide positions relative to HXB2 5197-5219 and IN4764AS (5' CCATTTGTACTGCTGTCTTAA 3’ 4764-4744)”.

2. The phylogenetic tree, figure 4, should contain all 88 sequences generated from sequencing using the two methods (50 from the In-house method and 38 from VS.

Response:

Thank you for highlighting this. The phylogenetic tree figure 4, is now figure 3. We have included a phylogenetic tree including all the sequences and a reference strain.

In addition, this has been included in the manuscript from line 222; “The 88 integrase sequences generated by the VS-Int and IH-Int methods were then compared for quality assurance and clustering using molecular phylogenetic analysis by maximum likelihood method in MEGA 7 software (39). The Phylogenetic tree was constructed using Tamura-Nei substitution model with gamma distribution rates among sites and inferred from 1000 bootstrap replicates (fig3) (40)”.

3. Figure 3 is not clear

Response: 

Fig 3 has been renamed fig 2. The figure has been reattached with a better resolution. 

4.All the sequences above should also be submitted to Genbank instead of just the 33 pairs.

Response: 

The 33 paired sequences have been submitted to Genbank and the accession numbers are available and are included in line 247 under materials and methods and subheading accession numbers; “The 33 pairs of nucleotide sequences obtained in our study were submitted to national center for biotechnology information (NCBI) GenBank and their accession numbers are MN037428 to MN037493”. 

We are in process of submitting to GenBank the remaining sequences.

5. Please the comments on the attached file for the rest of the comments.

Response: 

In addition, during revision of table 2 with our data set, we identified some minor corrections; under the IH-Int column, row 3, Q3 is 4.8 not 4.9. Row 4, percentage is 3.6% not 3.8%, and row 5 Q1;Q3 are 2.6 and 2.7 instead of 2.4,2.8.

Under the VS-Int column, row 2 Q3 is 4.8 not 4.9, row 3 Q1;Q2 is 2.8,4.5 not 2.6,5.0 and row 5 Q1,Q3 is 2.7,2.8 not 2.6,2.9.

---

## [Decision Letter · Decision Letter 1]

10 Oct 2019

Comparison of an in-house ‘home-brew’ and commercial ViroSeq integrase genotyping assays on HIV-1 subtype C samples

PONE-D-19-15741R1

Dear Dr. Gaseitsiwe,

We are pleased to inform you that your manuscript has been judged scientifically suitable for publication and will be formally accepted for publication once it complies with all outstanding technical requirements. Please note the two comments from reviewer #2 - it would be appreciated if you could make those minor edits to the manuscript. 

With kind regards,

Richard John Lessells, BSc, MBChB, MRCP, DTM&H, DipHIVMed, PhD

Academic Editor

PLOS ONE

Additional Editor Comments (optional):

Reviewers' comments:

Reviewer's Responses to Questions

**Comments to the Author**

1. If the authors have adequately addressed your comments raised in a previous round of review and you feel that this manuscript is now acceptable for publication, you may indicate that here to bypass the “Comments to the Author” section, enter your conflict of interest statement in the “Confidential to Editor” section, and submit your "Accept" recommendation.

Reviewer #1: All comments have been addressed

Reviewer #2: All comments have been addressed

2. Is the manuscript technically sound, and do the data support the conclusions?

Reviewer #1: Yes

Reviewer #2: Yes

3. Has the statistical analysis been performed appropriately and rigorously? 

Reviewer #1: Yes

Reviewer #2: Yes

4. Have the authors made all data underlying the findings in their manuscript fully available?

Reviewer #1: Yes

Reviewer #2: Yes

5. Is the manuscript presented in an intelligible fashion and written in standard English?

Reviewer #1: Yes

Reviewer #2: Yes

6. Review Comments to the Author

Reviewer #1: I have reviewed all responses provided by the authors. All comments have been adequately addressed and met my expectations.

Reviewer #2: Thank you for sufficiently addressing the comments by the reviewers.

Minor comments:

1. When citing the HIVDB algorithm used for the interpretation of of HIVDR data from sequence data please use the recommended citation (Liu TF, Shafer RW(2006). Web Resources for HIV type 1 Genotypic-Resistance Test Interpretation. Clin Infect Dis 42(11):1608-18. Epub 2006 Apr 28). PLEASE see the following link on how to cite, "" ext-link-type="uri" xlink:type="simple">https://hivdb.stanford.edu/pages/citation.html"

2. Since you analyzing sequence data not mutations, for the HIVDB weblink you should use "" ext-link-type="uri" xlink:type="simple">https://hivdb.stanford.edu/hivdb/by-sequences/" INSTEAD of "" ext-link-type="uri" xlink:type="simple">https://hivdb.stanford.edu/hivdb/by-mutations/"

7. PLOS authors have the option to publish the peer review history of their article (what does this mean?). If published, this will include your full peer review and any attached files.

Reviewer #1: No

Reviewer #2: Yes: Justen Manasa

---

## [Editor Report · Acceptance letter]

30 Oct 2019

PONE-D-19-15741R1 

Comparison of an in-house ‘home-brew’ and commercial ViroSeq integrase genotyping assays on HIV-1 subtype C samples 

Dear Dr. Gaseitsiwe:

I am pleased to inform you that your manuscript has been deemed suitable for publication in PLOS ONE. Congratulations! Your manuscript is now with our production department. 

With kind regards,

on behalf of

Dr. Richard John Lessells 

Academic Editor

PLOS ONE